# The *YBR056W-A* and Its Ortholog *YDR034W-B* of *S. cerevisiae* Belonging to CYSTM Family Participate in Manganese Stress Overcoming

**DOI:** 10.3390/genes14050987

**Published:** 2023-04-27

**Authors:** Anton Zvonarev, Larisa Ledova, Lubov Ryazanova, Airat Valiakhmetov, Vasilina Farofonova, Tatiana Kulakovskaya

**Affiliations:** 1Federal Research Center “Pushchino Scientific Center for Biological Research of the Russian Academy of Sciences”, Skryabin Institute of Biochemistry and Physiology of Microorganisms, 142290 Pushchino, Russialedova@ibpm.ru (L.L.);; 2Federal Research Center “Pushchino Scientific Center for Biological Research of the Russian Academy of Sciences”, Institute for Biological Instrumentation of the Russian Academy of Sciences, 142290 Pushchino, Russia

**Keywords:** CYSTM, manganese, stress, *MNC1*, *YDR034W-B*, yeast

## Abstract

The CYSTM (cysteine-rich transmembrane module) protein family comprises small molecular cysteine-rich tail-anchored membrane proteins found in many eukaryotes. The *Saccharomyces cerevisiae* strains carrying the CYSTM genes *YDRO34W-B* and *YBR056W-A* (*MNC1*) fused with *GFP* were used to test the expression of these genes under different stresses. The *YBR056W-A* (*MNC1*) and *YDR034W-B* genes are expressed under stress conditions caused by the toxic concentrations of heavy metal ions, such as manganese, cobalt, nickel, zinc, cuprum, and 2.4-dinitrophenol uncoupler. The expression level of *YDR034W-B* was higher than that of *YBR056W-A* under alkali and cadmium stresses. The Ydr034w-b-GFP and Ybr056w-a-GFP proteins differ in the cellular localization: Ydr034w-b-GFP was mainly observed in the plasma membrane and vacuolar membrane, while Ybr056w-a-GFP was observed in the cytoplasm, probably in intracellular membranes. The null-mutants in both genes demonstrated decreased cell concentration and lytic phenotype when cultivated in the presence of excess manganese. This allows for speculations about the involvement of Mnc1 and Ydr034w-b proteins in manganese stress overcoming.

## 1. Introduction

Yeast cells acquire a variety of stress-tolerant mechanisms, including the induction of stress proteins via stress-triggered signal-transduction pathways [1,2,3,4,5]. The proteins of the CYSTM superfamily were proposed to be a part of cellular protective mechanisms against stresses in yeast [6]. Multiple sequence alignment revealed a cysteine-rich transmembrane module, CYSTM, in a wide range of small molecular tail-anchored membrane proteins in eukaryotes, including humans [6]. To date, the proteins of the family have not been fully annotated. The *CYSTM1* gene has been identified in mammals; however, its function is still unclear (https://www.ebi.ac.uk/interpro/entry/InterPro/IPR043240/, accessed on 24 April 2023). The protein encoded by the *CYSTM1* gene has been shown to be a biomarker of Huntington’s disease [7].

The functions of proteins belonging to the CYSTM superfamily are best studied in plants. The significance of many CYSTM proteins in stress overcoming in plant cells has been demonstrated [8,9,10,11,12,13]. Some of the plant CYSTM proteins confer tolerance to cadmium and copper in *Digitaria ciliaris* and *Oryza sativa* [8]. Heterologous expression in yeast of plant *CDT1* belonging to this group conferred metal resistance by preventing the uptake of metal ions into the yeast cell [8]. The Arabidopsis representative of the superfamily, *PCC1*, was induced via the salicylic acid–dependent pathway upon pathogen exposure, and its overexpression leads to increased resistance to oomycetes [9]. In *Arabidopsis thaliana*, 13 CYSTM genes have been identified; they show different expression levels in various tissues, at different stages of development and under various stresses [10,11]. It suggests the diversity of their functions [10,11]. For example, the transcription of *CYSTM4*, *CYSTM5,* and *CYSTM6* was suppressed in shoots, but induced in roots under heat stress. Their expression was also induced by cold, polyethylene glycol, or salt in the root and shoot. *CYSTM2*, *CYSTM3*, and *CYSTM7* were upregulated by salt, cold, or drought treatment [10]. The involvement of *CYSTM3* in Na^+^ homeostasis was proposed [10]. The transcript of one gene (*At1g05340*) of *A. thaliana* encoding a CYSTM protein is induced mainly by heat and to a lesser extent by UV, but less by NaCl or sorbitol [12]. Functional analysis of *At1g05340* and its paralog *At2g32210* using T-DNA insertional mutants revealed a decrease in seedlings root length, an increase in sensitivity to heat stress, and to a lesser extent to UV stress, compared with the effect on wild-type plants [12]. The sensitivity of these mutants to salt or osmotic stresses did not differ from wild-type response, indicating a specific function for these genes in heat and UV stresses [12]. Heat and UV increased reactive oxygen species levels in the wild type; however, their levels were higher in the mutant line than in the wild type subjected to heat treatment but were similar in the mutant lines and wild type subjected to UV stress [12]. The results suggest that these CYSTM proteins are necessary for thermotolerance and protection from UV exposure; the proteins, most likely, act in heat stress by reducing reactive oxygen species level by yet unknown mechanism [12]. The overexpression of pathogen-induced cysteine-rich transmembrane proteins in *A. thaliana* enhances resistance against pathogens and stimulates hypocotyl growth, suggesting their potential role in both processes [13]. The CYSTM peptides in *A. thaliana* displayed various subcellular localization, and most of them were detected in the plasma membrane [10,13] and in the cytoplasm [10].

Due to the highly conservative structure of the CYSTM module, it was assumed that a role in stress response and, more specifically, in providing resistance to damaging substances might be a general function of the superfamily [6]. The conserved acidic residue could allow tight association with certain types of lipids and alter the permeability of the plasma membrane to damaging substances [6]. The changes in the plasma membrane could explain the role of CYSTM proteins in plant pathogen resistance, probably by blocking the invasion of intracellular pathogens. The peculiar arrangement of sulfhydryl groups within the membrane could also alter the redox potential of the membrane or potentially directly chelate metal ions [6]. Data on the functions of CYSTM proteins in plants indicate that their various representatives may be involved in adaptation to different types of stress.

In *S. cerevisiae* several CYSTM proteins were revealed (Table 1). In the SGD database (https://www.yeastgenome.org/, accessed on 10 March 2023) the CYSTM proteins are mentioned mainly as unannotated ones. These proteins demonstrate similarity to the CYSTM proteins of other eukaryotes (Figure 1). The features found in the structure of CYSTM yeast proteins are widely conserved throughout the family and found in no other membrane protein families. These features include an N-terminal cytoplasmic element that is predicted to adopt a β-strand connected by a highly variable linker to a C-terminal TM helix with 5–6 cysteine residue followed by an acidic residue [6]. The 3–4 cysteine residues occur consecutively and constitute a conserved cysteine site that is characteristic of this superfamily [6]. The article analyzing the structural features of the CYSTM superfamily describes four representatives revealed in yeast: *YDL012C*, *YDR210W*, *YBR016W*, and *YDR034W-B* [6]. The suggestion that they are tail-anchored membrane proteins has been experimentally confirmed [14,15].

The yeast co-expression network analysis suggests that the proteins *YDL012C*, *YDR210W*, and *YBR016W* are together involved in resistance against the DNA-damaging agents mitomycin C, the replication inhibitor methotrexate, the oxidizing agent hydrogen peroxide, and the membrane destabilizing agent 1,8-nonadiene [6].

*YDL012C* and *YDR210W* were shown to overlap in the chemicals against which they provide resistance, suggesting that these proteins might function together as a complex [6]. *YDL012C* and *YBR016W* target GFP in the plasma membrane [15]. Both *YDL012C* and *YBR016W* encoded proteins are localized mostly in regions of new membrane synthesis, toward the emergent buds of dividing cells [15].

*YDR034W-B* has a paralog, *YBR056W-A*. DNA microarray analysis made it possible to reveal a global set of yeast genes induced and repressed in response to various stresses, including variations in temperature, oxidation, nutrients, pH, and osmolarity, which indicates that nearly half of the genome is involved in the responses to environmental changes [5]. Among these genes, the so-called common environmental response (CER) genes were identified; their expression changed under a variety of stresses, including temperature shift, peroxide, osmotic and pH stresses [5]. One of the CER genes with its expression increasing under the above stress conditions was *YBR056W-A* [5]. The enhanced expression of *YBR056W-A (MNC1*) in *S. cerevisiae* cells adapted to growth in the presence of 2 mM Mn^2+^ and the lytic phenotype of the *∆ybr056w-a* strain in the presence of excess manganese have been shown, suggesting the possible role of this protein in manganese stress overcoming [16]. As for *YDR034W-B* encoded protein, there is no information about its role in stress conditions. Under normal growth conditions, the expression of this protein is negligible similar to *YBR056W-A* encoded protein [16].

This study aimed to compare the stress conditions in which the expression of *YDR034W-B* and *YBR056W-A* is manifested and to assess the localization of GFP-fused proteins in the cells of *S. cerevisiae*. The effect of knockout mutations in *YDR034W-B* and *YBR056W-A* on resistance to stress caused by heavy metal ions was also assessed.

## 2. Materials and Methods

### 2.1. Yeast Strains and Cultivation Conditions

The *S. cerevisiae* wild-type (WT) strain BY4742 (*MATα his3*Δ*1 leu2*Δ*0 lys2*Δ*0 ura3*Δ*0*) and BY4742-derived mutant strains *(∆ydro34w-b* and *∆ybro56w-a*) were obtained from the Euroscarf collection.

The *S. cerevisiae* GFP fusion strains *YDRO34W-B-GFP* and *YBR056W-A-GFP* derived from the BY4741 (*MATa his3*Δ*1 leu2*Δ*0 lys2*Δ*0 ura3*Δ*0*) strain were obtained from the Dharmacon collection. The strains were maintained on solid YPD medium containing 2% glucose, 2% peptone (Pronadisa, Madrid, Spain), and 1% yeast extract (Pronadisa, Madrid, Spain) supplemented with 2% agar (Sigma-Aldrich, Sigma-Aldrich, St. Louis, MO, USA). The start cultures were cultivated in 100 mL of the same YPD medium in Erlenmeyer flasks at 28 °C and 145 rpm for 24 h.

### 2.2. Fluorescence Microscopy

To test the effect stress factors on the expression of GFP fusion proteins, the cells of GFP fusion strains were cultivated in the control YPD medium and YPD media supplemented with one of the following components: 40 mM KOH, 0.2 mM Cd(CH_3_COO)_2_·2 H_2_O, 2 mM MnSO_4_·4H_2_O, 2 mM CoSO_4_·7H_2_O, 2 mM NiSO_4_·7H_2_O, 2 mM ZnSO_4_·7H_2_O, 2 mM CuSO_4_·5H_2_O, or 2 mM H_2_O_2_. The cultivation was performed in Erlenmeyer flasks in 100 mL of YPD at 28 °C and 145 rpm for 24 h.

To test the effect of 2,4-dinitrophenol, manganese, and peroxide on GFP fusion proteins expression during a short incubation time, the cells of the *YDRO34W-B-GFP* and *YBR056W-A-GFP* strains were cultivated in 2 mL of YPD with stirring for 24 h to a culture absorption of 20 (measured in a 1-cm cuvette at 600 nm). After cultivation, the medium was supplemented with one of the following components: 0.2 mM 2,4-dinitrophenol (Sigma, St. Louis, MO, USA), or 5 mM MnSO_4_·4H_2_O, or 2 mM H_2_O_2_, and the cultivation was continued for 0.5, 1 and 1.5 h. The control cultivation was performed for both strains under the same conditions in YPD.

After cultivation, the cells were examined in an AXIO Imager A1 fluorescent microscope (Zeiss, Jena, Germany) with a 56HE filter set (Zeiss) at a wavelength of 480 nm (maximum excitation) and 512–630 (emission). Images were obtained with Axiocam 506 (Zeiss).

### 2.3. Staining with Nile Red

The cells of *YDRO34W-B-GFP* and *YBR056W-A-GFP* strains were cultivated in YPD supplemented with 4 mM Mn^2+^ for 24 h. For visualization of lipids, the living cells were stained with Nile red (N1142, Thermo Scientific, Waltham, MA, USA) [17]. The cells were washed once in 0.025 M Hepes-KOH, pH 7.0, then incubated for 10 min at room temperature in the same buffer supplemented with the Nile red (the stock solution containing 1 mg/mL of Nile red in ethanol was diluted l:100 with Hepes-KOH, pH 7.0).

The cells were examined by phase-contrast and fluorescent microscopy in an AXIO Imager A1 (Zeiss, Jena, Germany) with a filter set of 56HE (Zeiss, Jena, Germany) at a wavelength of 450–500 nm for excitation and 512 + 630 nm for emission. An Axiocam 506 camera (Zeiss, Jena, Germany) was used to obtain images.

### 2.4. Determination of Effects of Different Concentrations of Heavy Metal Ions

The GFP strains *YDRO34W-B-GFP* and *YBR056W-A-GFP* were used to assess the effects of different concentrations of cadmium and manganese ions on GFP protein fluorescence intensity and fluorescence cell count.

The *S. cerevisiae* wild-type (WT) strain BY4742 and BY4742-derived mutant strains (*∆ydro34w-b* and *∆ybro56w-a*) were used to assess the effects of different concentrations of cadmium and manganese ions on cell growth.

The yeast strains were cultivated in a liquid YPD medium in sterile multi-well plates at 28 °C and 400 rpm for 24 h in a thermoshaker. Yeast samples were added to the normalized initial cell concentration (0.1 × 10^8^ cell/ mL) to the wells containing 0.2 mL of YPD medium supplemented with different concentrations of Cd(CH_3_COO)_2_·2H_2_O or MnSO_4_·4H_2_O. After cultivation, the cell concentration in culture samples was measured with a NovoCyte Flow cytometer. The samples were also examined with an AXIO Imager A1 ZEISS microscope (Oberkochen, Germany).

### 2.5. Flow Cytometry

The cell concentration in the cultivation medium was determined by flow cytometry. After a series of dilutions of the cell suspension with water, the number of cells in 25 μL in each sample was counted on a NovoCyte Flow cytometer (Agilent, Santa Clara, CA, USA). Expression of GFP-tagged proteins was determined on a NovoCyte flow cytometer using 488 nm for excitation and 585 nm for emission. A total of 100,000 cells were counted at each experimental point. All assays were repeated 4–5 times, and the mean results are presented.

### 2.6. Statistics

The experiments were performed in triplicate, and the results are presented as the mean with standard deviation (Excel). Statistical analysis was performed by Excel using Student’s t-test. The typical and most representative micrographs selected from 10–20 images obtained in independent experiments are presented.

## 3. Results

The *S. cerevisiae* strains *YDRO34W-B-GFP* and *YBR056W-A-GFP* carrying the *YDRO34W-B* and *YBR056W-A (MNC1)* genes fused with GFP were used to test the expression of the above genes in the presence of heavy metal ions, alkali, and H_2_O_2_ by fluorescence microscopy. The cells were cultivated for 24 h. Stress factor concentrations were chosen based on our previous work with related strain BY4741 [18]. The cells grown in the control YPD medium demonstrated negligible expression of both proteins (Figure 2). No expression of these proteins was observed upon the addition of 2 mM H_2_O_2_ to the YPD medium, the picture was the same as control (Figure 2). Green fluorescence of the cells indicating the enhanced expression of both *MNC1* and *YDRO34W-B* was observed in the cells grown in YPD supplemented with Mn^2+^ or Co^2+^ or Ni^2+^ or Zn^2+^ or Cu^2+^ (Figure 2). In the medium with KOH or Cd^2+^, the fluorescence of *YDRO34W-B-GFP* cells was much more pronounced compared with the fluorescence of *YBR056W-A-GFP* cells (Figure 2).

For quantification, fluorescence levels at a wavelength of 585 nm were compared using flow cytometry of cells grown in the presence of manganese and cadmium ions as an example. The data obtained (Appendix A) confirmed that the fluorescence of the cells of both strains grown in the presence of 4 mM Mn^2+^ was at a similar level. When the cells were grown in the presence of 0.2 mM Cd^2+^, the fluorescence level of cells of the *YDRO34W-B-GFP* strain was significantly higher than that of the *YBR056W-A-GFP* strain (Appendix A).

We also assessed the dependence of the proportion of fluorescing cells on concentrations of cadmium and manganese by flow cytometry. In the case of Cd^2+^, the proportion of fluorescing cells was higher for the *YDRO34W-B-GFP* strain, and in the case of Mn^2+^, this proportion was higher for the cells of the *YBR056W-A-GFP* strain (Figure 3). This was observed over the entire range of used concentrations of metal ions.

Thus, we have observed differences in the expression of Ydro34w-b and Mnc1 proteins depending on the type of stress.

The localization of CYSTM proteins in membranes, including the cytoplasmic membrane, was predicted due to the analysis of their structure [6] and was experimentally confirmed for some yeast proteins belonging to this family [14,15]. Our microscopic data do not contradict this view (Figure 2).

To improve the visualization of membrane structures, we used Nile red to stain the cells grown in the presence of 4 mM Mn^2+^ (Figure 4).

Because excitation of the Nile red starts at a wavelength of 410 nm, we can contemporaneously observe both lipids and GFP proteins in yeast cells using a Zeiss 56HE filter kit (a wavelength of 450–500 nm for excitation and 512 + 630 nm for emission). The fluorochrome stained vacuoles, but the plasma membrane was not visualized because of a high level of GFP fluorescence. Nevertheless, the obtained microphotographs show that in the cells of the *YDRO34W-B-GFP* strain, the GFP-fused protein is localized in the periphery of the cell and the vacuolar membrane, while in the cells of the *YBR056W-A-GFP* strain, the GFP-fused protein is localized in the cytoplasm, without concentrating in the cell periphery or vacuole. This method did not allow for precise determination of the Mnc1protein localization. Taking into account the presence of a membrane domain, we speculate that Mnc1 can be localized in the membranes of the endoplasmic reticulum. The data obtained confirm the idea of membrane localization of these members of the CYSTM family and indicate differences in their localization.

Many types of stress lead to a decrease in the electrochemical gradient on the yeast plasma membrane, and this disturbance may be the primary step of the stress response [19,20]. In this regard, we have checked how the known uncoupler 2,4-dinitrophenol [21] affects the expression of *MNC1* and *YDRO34W-B*. The short-term incubation of the *YDRO34W-B-GFP* and *YBR056W-A-GFP* strains with 2,4-dinitrophenol, Mn^2+,^ and H_2_O_2_ was used for this purpose. No green fluorescence of the cells of both strains was observed in the control cells (Figure 5).

The fluorescence was not detected by fluorescence microscopy after 30 and 60 min of incubation with 2,4-dinitrophenol or Mn^2+^. Green fluorescence of the cells of both strains appeared in the case of cultivation for 1.5 h in the presence of 2,4-dinitrophenol and Mn^2+^ (Figure 5), but not H_2_O_2_. The cell demonstrated no fluorescence in the case of H_2_O_2_ (the picture was the same as control, Figure 5). Hence, 2,4-dinitrophenol stimulated the expression of both genes similarly to Mn^2+^, and we suggest that the disturbance of the electrochemical gradient on the plasma membrane may serve as a signal of this stimulation.

We have also checked the effects of the knockout mutations in *MNC1* and *YDRO34W-B* on the growth and cell morphology under Cd^2+^- and Mn^2+^-induced stresses. The changes in cell concentrations (Figure 6a) and cellular morphology (Figure 6c) in the WT-type strain and *∆mnc1* and *∆ydro34w-b* strains under Cd^2+^ stress were similar. In the presence of Mn^2+^ (2–4 mM), cell concentration decreased for the *∆mnc1* and *∆ydro34w-b* strains (Figure 6b). Cell lysis was also observed by light microscopy (Figure 6c). The effect of knockout mutations was not significant but it suggested the possibility of the participation of the studied proteins in overcoming stress caused by a toxic Mn^2+^ concentration.

## 4. Discussion

The role of CYSTM proteins in metal homeostasis was proposed long ago: the arrangement of sulfhydryl groups of the protein within the membrane could alter the redox potential of the membrane or directly chelate metal ions [6]. The enhanced expression of *MNC1* but not *YDR034W-B* was observed in *S. cerevisiae* cells adapted to the growth at toxic manganese concentrations [16]. In this article, the transcriptome of cells of the stationary growth stage has been analyzed (120-h cultivation in the presence of 2.5 mM manganese salt); in this study, we have analyzed the cells of the early active-growth stage (24-h cultivation in the presence of 2.5 mM manganese salt). Possibly, the observed difference in the *YDR034W-B* expression is associated with the growth stage. Previously we suggested the plasma membrane localization of Mnc1 [16] based on data on the structure and localization of other CYSTM proteins [6,15]. The data obtained for the strains containing the GFP-fusion gene in the present work indicate that Mnc1 is localized mainly in the cytoplasm, probably in intracellular membranes.

The expression of *YDR034W-B* and *MNC1* in the cells grown in the control YPD medium was very low [16]. In this work, we have first identified the stress conditions in which *YDR034W-B* gene expression increased. This expression increased in yeast cells cultivated in the presence of toxic concentrations of Mn^2+^, Co^2+^, Ni^2+^, Cu^2+^, Zn^2+^, Cd^2+^, or KOH. The *MNC1* gene is expressed in the presence of toxic concentrations of Mn^2+^, Co^2+^, Ni^2+^, Cu^2+^, or Zn^2+^, while its expression shows a lower level in the presence of Cd^2+^ or KOH.

To our surprise, we did not observe the expression of these proteins under peroxide stress, although an increase in the expression of *YBR056W-A* under this type of stress has been previously reported [6]. Perhaps, this is due to differences in cultivation conditions and stress exposure. We did not observe any effect on resistance to cadmium ions of knockout mutations in both genes studied. At the same time, we noted a decrease in the resistance of knockout mutant cells to manganese ions. Ions of cadmium [22,23] and manganese [24,25,26] significantly differ in the mechanism of toxic action, the range of toxic concentrations, and the biological role. Cadmium does not participate in metabolic processes in yeast, while manganese is an essential trace element for yeast, being a cofactor for numerous metalloenzymes [25] and a necessary component of the antioxidant system [26]. In this regard, the system of manganese homeostasis in yeast includes a number of membrane proteins that differ in the mechanism of action and cellular localization. We speculate that Mnc1 and Ydro34w-b proteins may be components of the manganese homeostasis system in yeast cells. We propose *MNC2* (manganese-chelating protein 2) as a possible gene name for *YDRO34W-B*.

The significance of studying the yeast manganese homeostasis system is summarized in the review, where the authors indicate the functional conservation of proteins involved in manganese homeostasis from yeast to humans [25]. This fact makes yeast a relevant model to find out new aspects of human neurodegenerative diseases and to propose new therapeutic approaches to their treatment [25]. It is noteworthy that one of the markers of Hadington’s disease is a protein of the CYSTM family [7]. Thus, the study of CYSTM proteins is of interest for further research on stress adaptation in eukaryotes.

## Figures and Tables

**Figure 1 genes-14-00987-f001:**
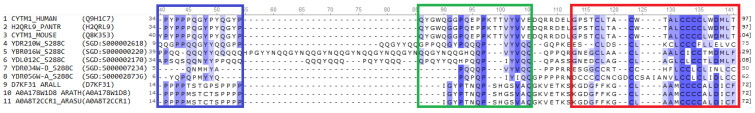
The sequence alignment of some proteins of CYSTM domain superfamily. The alignment of protein sequences was performed by UGENE (http://ugene.net/ru/, accessed on 10 March 2023) using the ClustalW algorithm. The columns are colored according to the similarities, and the CYSTM domain is indicated in red. The sequences are labeled using the gene names and species abbreviations, the database sequence codes are indicated in parentheses. Sequences 1–3 and 9–11 are from UniProt (https://www.uniprot.org/, accessed on 10 March 2023), and sequences 4–8 are from SGD (https://www.yeastgenome.org/, accessed on 10 March 2023). The species abbreviations are—HUMAN: *Homo sapiens*; PANTR: *Pan troglodytes*; MOUSE: *Mus musculus*; S288C: *S. cerevisiae* (S288C strain); ARALL: *Arabidopsis lyrata*; ARATH: *A. thaliana*; ARASU: *Arabidopsis suecica*.

**Figure 2 genes-14-00987-f002:**
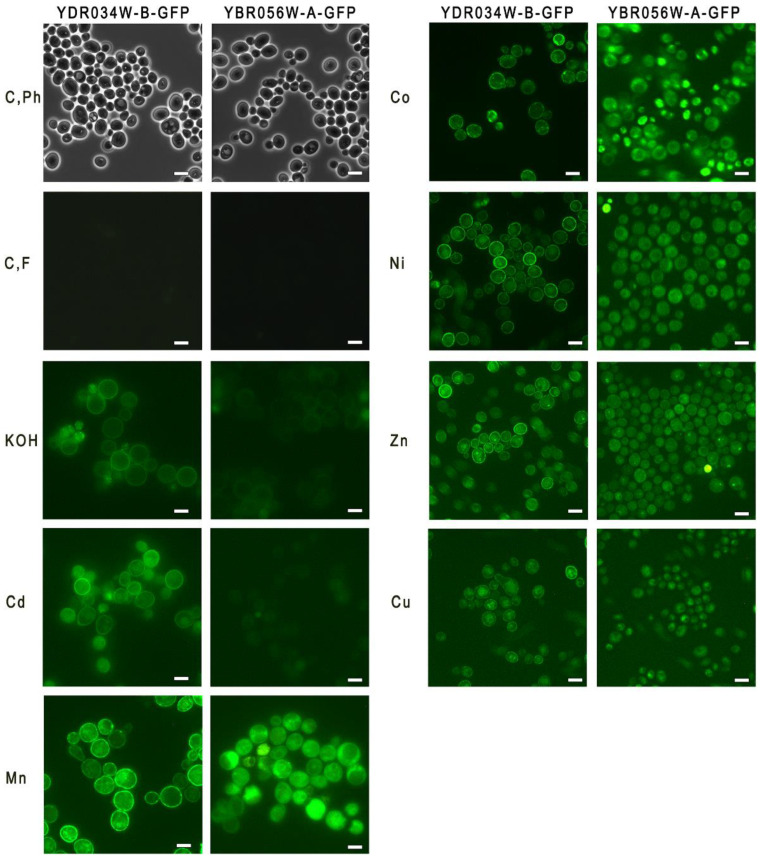
Micrographs of cells of strains *YBR056W-A-GFP* and *YDR034W-B-GFP*. The cells were cultivated in YPD medium for 24 h. C, Ph—phase contrast microscopy, control cultivation; C, F—fluorescence microscopy, control cultivation; KOH—fluorescence microscopy, cultivation in the presence of 40 mM KOH, Cd—fluorescence microscopy, cultivation in the presence of 0.2 mM Cd^2+^; Mn, Co, Ni, Zn, Cu—fluorescence microscopy, cultivation in the presence of 2 mM of Mn^2+^, Co^2+^, Ni^2+^, Zn^2+^, Cu^2+^, respectively. The bar line is 5 μm.

**Figure 3 genes-14-00987-f003:**
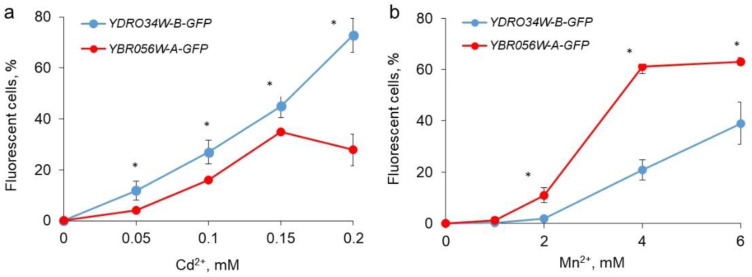
The effects of different concentrations of Cd^2+^ (**a**) and Mn^2+^ (**b**) on the proportion of green fluorescence cells in cell populations of *S. cerevisiae* strains *YBR056W-A-GFP* and *YDR034W-B-GFP*. The yeast was cultivated for 24 h as described in the Materials and Methods Section and counted by flow cytometer. The experiments were performed in triplicate; the values denote mean, the whiskers denote s.d. * *p* < 0.01, and significance was assessed with the two-tailed Student’s t-test (*YDR034W-B-GFP* vs. *YBR056W-A-GFP*).

**Figure 4 genes-14-00987-f004:**
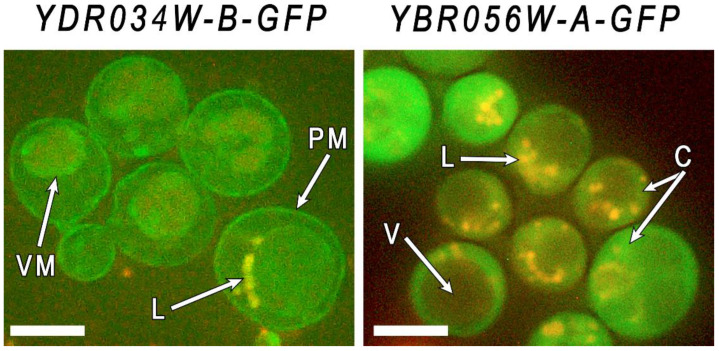
Micrographs of cells of strains *YBR056W-A-GFP* and *YDR034W-B-GFP* grown in YPD in the presence of 4 mM of Mn^2+^ for 24 h. Staining with Nile red. PM—plasma membrane, VM—vacuolar membrane, L—lipid inclusions, V—vacuole, C—cytoplasm. The bar line is 5 μm.

**Figure 5 genes-14-00987-f005:**
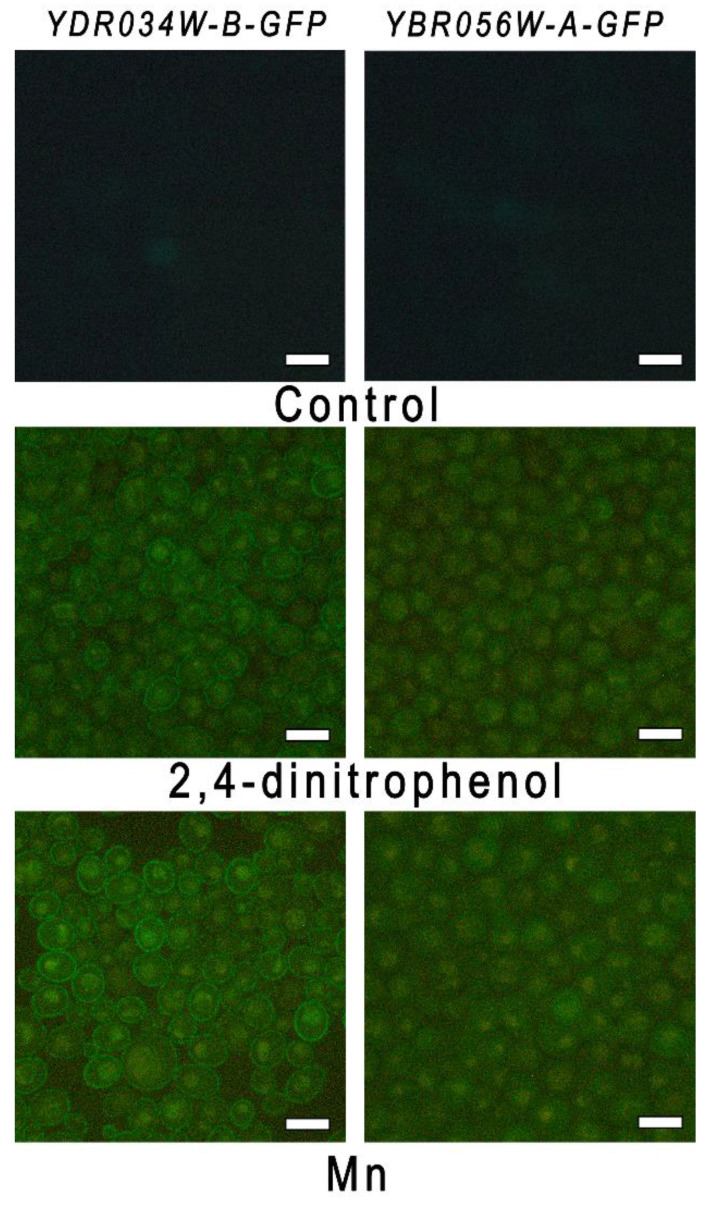
Micrographs of cells of strains *YBR056W-A-GFP* and *YDR034W-B-GFP* obtained by fluorescence microscopy. The cells were incubated in control YPD and in the presence of 0.2 mM dinitrophenol or 5 mM MnSO_4_ for 1.5 h. The bar line is 5 μm.

**Figure 6 genes-14-00987-f006:**
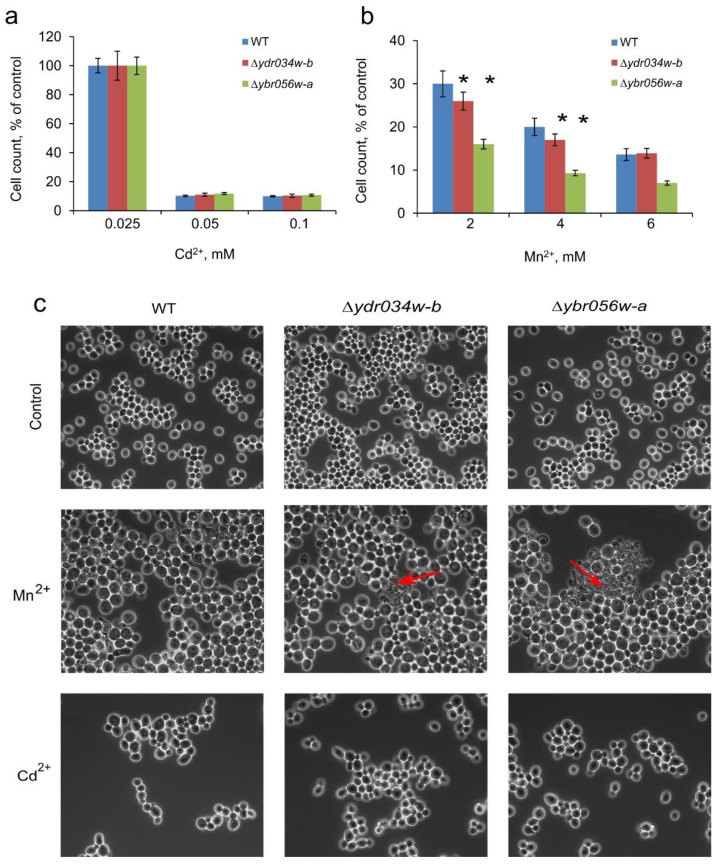
(**a**)—The cell concentration of *S. cerevisiae* wild-type (WT) strain BY4742 and BY4742-derived mutant strains *(∆ydr034w-b* and *∆ybro56w-a*); cultivation in control YPD and in YPD supplemented with various concentrations of Cd^2+^(**a**) or Mn^2+^ (**b**). The cell concentration of ~2.3 × 10^8^ cell/mL in control cultivation corresponds to 100%. The cells were cultivated for 24 h in multi-well plates. The experiments were performed in triplicate. The values denote mean, the whiskers denote s.d., * *p* < 0.05; in other cases the difference was insignificant, and the significance was assessed with the two-tailed Student’s t-test against WT. (**c**)—Phase-contrast micrographs of cells of wild-type (WT) strain BY4742 and BY4742-derived mutant strains *(∆ydr034w-b* and *∆ybro56w-a*) grown in YPD control medium and the presence of 2 mM Mn^2+^, or 0.075 mM Cd^2+^; red arrows indicate the agglomerates of lysed cells.

**Table 1 genes-14-00987-t001:** The CYSTM proteins of *S. cerevisiae* (https://www.yeastgenome.org/, accessed on 10 March 2023).

Systematic Gene Name	Molecular Mass of Protein, Da	Description
*YDR034W−B*	5968.2	Predicted tail-anchored plasma membrane protein; N- and C-terminal fusion proteins localize to the cell periphery and have a paralog, *YBR056W-A*, that arose from the whole genome duplication
*YBR056W-A* *(MNC1)*	7326.8	Putative membrane protein upregulated in toxic manganese levels
*YDL012C*	12,173.4	Tail-anchored plasma membrane protein, possibly involved in response to stress; has a paralog, *YBR016W*, that arose from the whole genome duplication
*YBR016W*	14,617.0	Tail-anchored plasma membrane protein; has similarity to hydrophilins, which are involved in the adaptive response to hyperosmotic conditions
*YDR210W*	8567.7	Predicted tail-anchored plasma membrane protein

## Data Availability

Not applicable.

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
