# Peer review of "The YBR056W-A and Its Ortholog YDR034W-B of S. cerevisiae Belonging to CYSTM Family Participate in Manganese Stress Overcoming"

_genes, 2023, doi:10.3390/genes14050987_

Round 1
Reviewer 1 Report
In this manuscript, Zvonarev and co-workers describe the role of two yeast genes belonging to the CYSTM family - YBR056w-A and YDR034w-B – in cell response to heavy metal stress, with a focus on manganese stress.
The study may be of interest in elucidating the role of CYSTM proteins in heavy metal homeostasis. Nevertheless, there are some issues that the authors need to address.
- - The rationale for using fixed concentrations of the ions tested is not fully clear. It is advised that the authors make concentration screenings for each metal, to determine the minimal concentrations that induce the gene expression. Also, for a concentration that induces a clear effect, it is important to monitor the onset of fluorescence as a function of metal exposure time.
- - The autors speculate that the role of Wdr034v-Bp and Wbr056w-Ap is to bind/coordinate the metal ions to the cell membranes. Further experiments are needed to support this hypothesis, for example monitoring of metal accumulation by WT and the corresponding knockout strains.
- - The pictures showing the localization of Wbr056w-Ap to internal membranes are not conclusive, although there are images in which localization at vacuolar membrane is apparent. Use of fluorescent organellar markers may be of help.
Author Response
Dear Reviewer 1,
Thank you very much for reviewing of our manuscript and fruitful comments. Based on your recommendation, we conducted a series of experiments and added the results to the manuscript. All changes in the manuscript are highlighted in yellow.
“The rationale for using fixed concentrations of the ions tested is not fully clear. It is advised that the authors make concentration screenings for each metal, to determine the minimal concentrations that induce the gene expression. Also, for a concentration that induces a clear effect, it is important to monitor the onset of fluorescence as a function of metal exposure time.”
- Stress factor concentrations were chosen based on our previous work using a related strain BY4741 (Tomashevsky A, Kulakovskaya E, Trilisenko L, Kulakovskiy IV, Kulakovskaya T, Fedorov A, Eldarov M. VTC4 Polyphosphate Polymerase Knockout Increases Stress Resistance of Saccharomyces cerevisiae Cells. Biology (Basel). 2021 May 30;10(6):487. doi: 10.3390/biology10060487.
For revised version of manuscript, we conducted the concentration screening for manganese and cadmium both for GFP fused strains (Figure 3) and knock out strains (Figure 6).
“The authors speculate that the role of Wdr034v-Bp and Wbr056w-Ap is to bind/coordinate the metal ions to the cell membranes. Further experiments are needed to support this hypothesis, for example monitoring of metal accumulation by WT and the corresponding knockout strains.”
- We agree that the data of our work are insufficient to suggest the possibility of the formation of complexes of the studied proteins with metal ions. This indeed requires further research. We changed the assumption on the role of these proteins in Abstract section: This allow to speculate the involvement of Mnc1 and YDR034W-B proteins in manganese stress overcoming.
“The pictures showing the localization of Wbr056w-Ap to internal membranes are not conclusive, although there are images in which localization at vacuolar membrane is apparent. Use of fluorescent organellar markers may be of help.”
- We agree with this comment and attempted to stain cells grown in the presence of manganese with FM4-64 (T13320, Thermo Scientific). However, these cells were fragile and lysed during the procedure recommended by the company manual. In a small number of preserved cells, it was not possible to observe the flurescence of intracellular membranes, and only numerous fluorescence inclusions were observed. We hypothesized that this is due to impaired endocytosis upon exposure to manganese, since this dye is known to enter yeast cells in this way (Best JT, Xu P, McGuire JG, Leahy SN, Graham TR. Yeast synaptobrevin, Snc1, engages distinct routes of postendocytic recycling mediated by a sorting nexin, Rcy1-COPI, and retromer Mol Biol Cell 2020;31(9):944-962 doi:10.1091/mbc.E19-05-0290). Therefore, special studies are needed to select the staining conditions by FM4-64 for yeast cells exposed to manganese.
To improve visualization of membrane structures, we stained the cells grown in the presence of 4 mM Mn2+ with Nile red (Figure 4). Due to the fact that excitation of the Nil red starts at a wavelength of 410 nm, we can immediately observe both lipids and GFP proteins in yeast cells using Zeiss 56HE filter kit (a wavelength of 450–500 nm for excitation and 512 + 630 nm for emission). The fluorochrome stained vacuoles, but plasma membrane was not visualized because of high level of GFP fluorescence. Nevertheless, the obtained microphotographs show that in the cells of YDRO34W-B-GFP strain the YDRO34W-B is localized in the periphery of the cell and in the vacuolar membrane, while in the cells of YBR056W-A-GFP strain Mnc1 protein is localized in the cytoplasm, without focusing on the cell periphery or vacuole. This method did not allow to precisely determine the localization of Mnc1. Taking into account the presence of a membrane domain, we speculate that Mnc1 can be localized in the membranes of the endoplasmic reticulum. The data obtained confirm the idea of membrane localization of these members CYSTM family and indicate differences in their localization.
We also changed the sentence in Abstract section: The YDR034W-B-GFP and YBR056W-A-GFP protein differ in the cellular localization: YDR034W-B-GFP was mainly observed in plasma membrane and vacuolar membrane, while YBR056W-A-GFP was observed in cytoplasm, probably in intracellular membranes.
Reviewer 2 Report
In the manuscript, yeast genes YDR034W-B and YBR056W-A, encoding cysteine-rich transmembrane proteins were studied. GFP fusions were used to reveal expression differences and subcellular localization. A mild phenotype for deletion strains ydr034w-b∆ and ybr056w-a∆ was observed but data are not fully convincing in the present form.
Almost all of the data shown are qualitative, not quantitative. Expression of YBR056W-A-GFP and YDR034W-B-GFP under different conditions was compared by microscopy. Pictures are presented without quantification but differences in expression are deduced. To support differences in GFP fusion protein levels, Western analysis is recommended (including a suitable loading control). Alternatively, or additionally, mRNAs of the two genes could be quantified by qPCR or Northern analysis.
The authors also state definitive localizations of the two proteins in the abstract (“YBR056W-A is localized in intracellular membranes, while YDR034W-B is localized in the cytoplasmic membrane”) which would require a proof of colocalization of GFP signal with stained cytoplasmic or vacuolar membrane. A staining with FM 4-64 is commonly used for this. In the current form, the pictures shown do not provide definite information about subcellular localization.
Figure 3 lacks a control, as it is it doesn’t support the conclusion that the uncoupler or Mn induces expression of YBR056W-A-GFP and YDR034W-B-GFP.
Cell concentrations in presence of Cd or Mn were determined with a flow cytometer using fluorescence activation (488-nm excitation and 585-nm emission was used, stated in section 2.3). However, in figure 4A (the only figure showing cell concentrations), deletion strains ydr034w-b∆ and ybr056w-a∆ were used, which should not show GFP fluorescence. It remains unclear how information on cell concentration was obtained.
The “Mn” part of figure 4A shows cell concentrations lower than 0.05x107 cells/ml, but according to the methods, inoculation of wells was at 0.5x107 cells/ml. If this low cell concentration compared to the control (reading around 4x107 cells/ml) was due to lysis, a massive effect would be expected to be observed in figure 4B, which isn’t the case. It appears that there was no growth at all for either the wild type or the mutants in the immunoplate cultures containing Mn, whereas the Mn free control cultures were able to grow. If the wild type was already completely inhibited under the conditions applied, no significant effect can be expected in the mutants and a variation of the Mn concentration should be considered to reach a point where the mutants clearly differ from the wild type. Since these observations are used to support a new phenotype (reduced cell concentration due to lysis) for the two genes, additional supportive evidence is strongly recommended. If there is enhanced cell lysis in the two mutants, this should also be observable by serial dilutions of O.D. 600 nm adjusted cultures followed by plate spot assays using plates containing or lacking Mn. Also, substantial cell lysis would result in differences in cell viability which can be determined by plating of cell dilutions, followed by colony counting.
Figure 4B shows few lysed cells in the selected pictures for the two deletion mutants, whereas no such lysed cells are seen in the control picture. These pictures are used to support the main conclusion of the manuscript (Title: YBR056W-A and its ortholog YDR034W-B … participate in manganese stress overcoming). To convincingly demonstrate a significant difference between the two mutants and the wild type, however, this effect would need quantification (independent cultures, counting of intact cells and counting of lysed cells, statistical significance test).
Author Response
Dear Reviewer 2,
Thank you very much for reviewing of our manuscript and fruitful comments. Based on your recommendation, we conducted a series of experiments and added the results to the manuscript. All changes in the manuscript are highlighted in yellow.
“Almost all of the data shown are qualitative, not quantitative. Expression of YBR056W-A-GFP and YDR034W-B-GFP under different conditions was compared by microscopy. Pictures are presented without quantification but differences in expression are deduced. To support differences in GFP fusion protein levels, Western analysis is recommended (including a suitable loading control). Alternatively, or additionally, mRNAs of the two genes could be quantified by qPCR or Northern analysis.”
- The studied proteins are mentioned in the SGD database as unannotated, they are not expressed under normal conditions. Since these are membrane proteins, their purification and production of antibodies is a separate experimental task; therefore, we are not able to use a Western blot at this stage of the study. Since they have a great similarity in structure, the selection of primers for PCR is also a special experimental problem. In this regard, we make quantitative measurements of the level of fluorescence and the number of fluorescent cells using flow cytometry and presented the results in Figure 3 and Supplementary Figure 1.
“The authors also state definitive localizations of the two proteins in the abstract (“YBR056W-A is localized in intracellular membranes, while YDR034W-B is localized in the cytoplasmic membrane”) which would require a proof of colocalization of GFP signal with stained cytoplasmic or vacuolar membrane. A staining with FM 4-64 is commonly used for this. In the current form, the pictures shown do not provide definite information about subcellular localization.”
- We agree with this comment and attempted to stain cells grown in the presence of manganese with FM4-64 (T13320, Thermo Scientific). However, these cells were fragile and lysed during the procedure recommended by the company manual. In a small number of preserved cells, it was not possible to observe the flurescence of intracellular membranes, and only numerous fluorescence inclusions were observed. We hypothesized that this is due to impaired endocytosis upon exposure to manganese, since this dye is known to enter yeast cells in this way (Best JT, Xu P, McGuire JG, Leahy SN, Graham TR. Yeast synaptobrevin, Snc1, engages distinct routes of postendocytic recycling mediated by a sorting nexin, Rcy1-COPI, and retromer Mol Biol Cell 2020;31(9):944-962 doi:10.1091/mbc.E19-05-0290). Therefore, special studies are needed to select the staining conditions by FM4-64 for yeast cells exposed to manganese.
To improve visualization of membrane structures, we stained the cells grown in the presence of 4 mM Mn2+ with Nile red (Figure 4). Due to the fact that excitation of the Nil red starts at a wavelength of 410 nm, we can immediately observe both lipids and GFP proteins in yeast cells using Zeiss 56HE filter kit (a wavelength of 450–500 nm for excitation and 512 + 630 nm for emission). The fluorochrome stained vacuoles, but plasma membrane was not visualized because of high level of GFP fluorescence. Nevertheless, the obtained microphotographs show that in the cells of YDRO34W-B-GFP strain the YDRO34W-B is localized in the periphery of the cell and in the vacuolar membrane, while in the cells of YBR056W-A-GFP strain Mnc1 protein is localized in the cytoplasm, without focusing on the cell periphery or vacuole. This method did not allow to precisely determine the localization of Mnc1. Taking into account the presence of a membrane domain, we speculate that Mnc1 can be localized in the membranes of the endoplasmic reticulum. The data obtained confirm the idea of membrane localization of these members CYSTM family and indicate differences in their localization.
We also changed the sentence in Abstract section: The YDR034W-B-GFP and YBR056W-A-GFP protein differ in the cellular localization: YDR034W-B-GFP was mainly observed in plasma membrane and vacuolar membrane, while YBR056W-A-GFP was observed in cytoplasm, probably in intracellular membranes.
“Figure 3 lacks a control, as it is it doesn’t support the conclusion that the uncoupler or Mn induces expression of YBR056W-A-GFP and YDR034W-B-GFP.”
- In the first version of the manuscript, we did not include photographs of control cells, since they are of little information due to the absence of green florescence. Now we have added a micrograph of control cells obtained with the same photographic parameters in Fig. 5 (former Figure 3).
“Cell concentrations in presence of Cd or Mn were determined with a flow cytometer using fluorescence activation (488-nm excitation and 585-nm emission was used, stated in section 2.3). However, in figure 4A (the only figure showing cell concentrations), deletion strains ydr034w-b∆ and ybr056w-a∆ were used, which should not show GFP fluorescence. It remains unclear how information on cell concentration was obtained.”
- The flow cytometer counts every event (cell passing through the laser beam), regardless of the presence of fluorescence. If the cells are fluorescent, the cytometer assigns a separate value (fluorescence intensity) to this event, which can be analyzed as a separate parameter. So, the cells which are not florescent (GFP fusion strains in control conditions or knock out mutant strains) can be count using the parameters indicated in Methods (see Supplementary figure 1).
“The “Mn” part of figure 4A shows cell concentrations lower than 0.05x107 cells/ml, but according to the methods, inoculation of wells was at 0.5x107 cells/ml. If this low cell concentration compared to the control (reading around 4x107 cells/ml) was due to lysis, a massive effect would be expected to be observed in figure 4B, which isn’t the case. It appears that there was no growth at all for either the wild type or the mutants in the immunoplate cultures containing Mn, whereas the Mn free control cultures were able to grow. If the wild type was already completely inhibited under the conditions applied, no significant effect can be expected in the mutants and a variation of the Mn concentration should be considered to reach a point where the mutants clearly differ from the wild type. Since these observations are used to support a new phenotype (reduced cell concentration due to lysis) for the two genes, additional supportive evidence is strongly recommended. If there is enhanced cell lysis in the two mutants, this should also be observable by serial dilutions of O.D. 600 nm adjusted cultures followed by plate spot assays using plates containing or lacking Mn. Also, substantial cell lysis would result in differences in cell viability which can be determined by plating of cell dilutions, followed by colony counting.”
- We apologize for the mistake in the indication of the cell concentration. In the revised manuscript, we present the results of a new experiment where we tested the effect of different concentrations of metals on cell concentration assayed by flow cytometry (Figure 6). In our previous work, we have found that the spot test cannot be applied effectively to manganese exposure (Andreeva, N., Kulakovskaya, E., Zvonarev, A., A Penin, A., Eliseeva, , Teterina, A., A Lando, A., Kulakovskiy, I.V., Kulakovskaya, T. Transcriptome profile of yeast reveals the essential role of PMA2 and uncharacterized gene YBR056W-A (MNC1) in adaptation to toxic manganese concentration. Metallomics 2017, 9, 175-182. doi: 10.1039/c6mt00210b. PMID: 28128390.
“Figure 4B shows few lysed cells in the selected pictures for the two deletion mutants, whereas no such lysed cells are seen in the control picture. These pictures are used to support the main conclusion of the manuscript (Title: YBR056W-A and its ortholog YDR034W-B … participate in manganese stress overcoming). To convincingly demonstrate a significant difference between the two mutants and the wild type, however, this effect would need quantification (independent cultures, counting of intact cells and counting of lysed cells, statistical significance test). “
- The micrograph in Fig. 6 (former Fig 4) is a typical picture, we chose it from more than 20 similar photos from several independent experiments. The statistical significance test was performed for data presented in Fig. 6 a and b.
Round 2
Reviewer 1 Report
The authors responded to most concerns raised by the reviewer.
Author Response
Dear Reviewer 1,
Thank you very much for careful reading of our manuscript.
With best regards,
Tatiana Kulakovskaya on behalf of all co-authors
Reviewer 2 Report
The authors have added quantitative and additional microscopy data which improved the manuscript. Also, statements about protein localization have been softened, which increased scientific quality. Some wording errors are present in the newly added text sections (marked in yellow). Please review before proceeding with publication.
Examples:
Line 24: “This allow to speculate the involvement of Mnc1 and Ydr034w- 24
b proteins in manganese stress overcoming.” Should read: “This allows to speculate about …”
Line 114: “YDL012C and YDR210W were shown to be overlap in the chemicals against they pro
vide resistance”. Should read “… were shown to overlap ..”
Line 333: “The MNC1 is expressed”. Should read “The MNC1 gene …”
Line 336: “To our surprise, we did not find the expression of these proteins…” Should read “ … we did not observe expression …”
Author Response
Dear Reviewer 2,
Thank you very much for careful reading of our manuscript.
We have made efforts to correct errors in our text according to your comments.
With best regards
Tatiana Kulakovskaya
on behalf of all co-authors